# SayNav: Grounding Large Language Models
# for Dynamic Planning to Navigation in New Environments

**Primary Keywords:** *Learning; Robotics; Knowledge Representation/Engineering*

## Abstract

Semantic reasoning and dynamic planning capabilities are crucial for an autonomous agent to perform complex navigation tasks in unknown environments. It requires a large amount of common-sense knowledge, that humans possess, to succeed in these tasks. We present SayNav, a new approach that leverages human knowledge from Large Language Models (LLMs) for efficient generalization to complex navigation tasks in unknown large-scale environments. SayNav uses a novel grounding mechanism, that incrementally builds a 3D scene graph of the explored environment as inputs to LLMs, for generating feasible and contextually appropriate high-level plans for navigation. The LLM-generated plan is then executed by a pre-trained low-level planner, that treats each planned step as a short-distance point-goal navigation sub-task. SayNav dynamically generates step-by-step instructions during navigation and continuously refines future steps based on newly perceived information. We evaluate SayNav on multi-object navigation (MultiON) task, that requires the agent to utilize a massive amount of human knowledge to efficiently search multiple different objects in an unknown environment. We also introduce a benchmark dataset for MultiON task employing ProcTHOR framework that provides large photo-realistic indoor environments with variety of objects. SayNav achieves state-of-the-art results and even outperforms an oracle based baseline with strong ground-truth assumptions by more than 8% in terms of success rate, highlighting its ability to generate dynamic plans for successfully locating objects in large-scale new environments.

## 1 Introduction

*Finding multiple target objects in a novel environment* is a relatively easy task for a human but a daunting task for an autonomous agent. Given such a task, humans are able to leverage common-sense priors like room layouts and plausible object placement to infer likely locations of objects. For example, there are higher chances of finding a pillow on the bed in the bedroom and a spoon on the dining table or in the kitchen. Humans are also capable of dynamically planning and adjusting their search strategies and actions based on new visual observations during exploration in a new environment. For example, a human would search for a spoon first instead of a pillow if entering a kitchen.

Such reasoning and dynamic planning capabilities are essential for an autonomous agent to accomplish complex navigation tasks in novel settings, such as searching and locating specific objects in new houses. However, current learning-based methods, with the most popular being deep reinforcement learning (DRL) (Anderson et al. 2018; Chaplot et al. 2020; Khandelwal et al. 2022), require massive amounts of training for the agent to achieve reasonable performance even for simpler navigation tasks, such as finding a single object (object-goal navigation) or reaching a single target point (point-goal navigation) (Anderson et al. 2018). Moreover, significant computational resources are needed to replicate human ability to generalize to new environments. Such computational demands impede the development of an autonomous agent to efficiently conduct complex tasks at unknown places.

In this paper, we propose **SayNav** – a new approach to leverage common-sense knowledge from Large Language Models (LLMs) for efficient generalization to complicated navigation tasks in unknown large-scale environments. Recently, agents equipped with LLM-based planners have shown remarkable capabilities to conduct complex manipulation tasks with only a few training samples (Ahn et al. 2022; Song et al. 2022). SayNav follows this trend of utilizing LLMs in developing generalist planning agents specifically for navigation tasks. To fully demonstrate and validate SayNav's capabilities, we choose a complex navigation task, multi-object navigation (MultiON). For this task, the agent needs to efficiently explore a new 3D environment to locate multiple different objects given the names of these objects. This task requires a large amount of prior knowledge and dynamic planning capabilities (similar to humans) for success.

MultiON task has emerged recently as a generalization of the Object-goal Navigation task. It was introduced by (Wani et al. 2020) as a task of "navigation to an ordered sequence of objects" and most of the other works on MultiON follow the same definition. (Gireesh et al. 2023) dropped the constraint of having a ordered sequence of the given objects and defined the goal as to localize certain number of objects in any order. This is how we also define our MultiON task as it poses much larger planning challenges and task complexities than the previous definition. Following the trend in MultiON task, we will be using three different objects as the targets for each episode for experiments. Note that SayNav is capable of searching for any number of objects in the en-

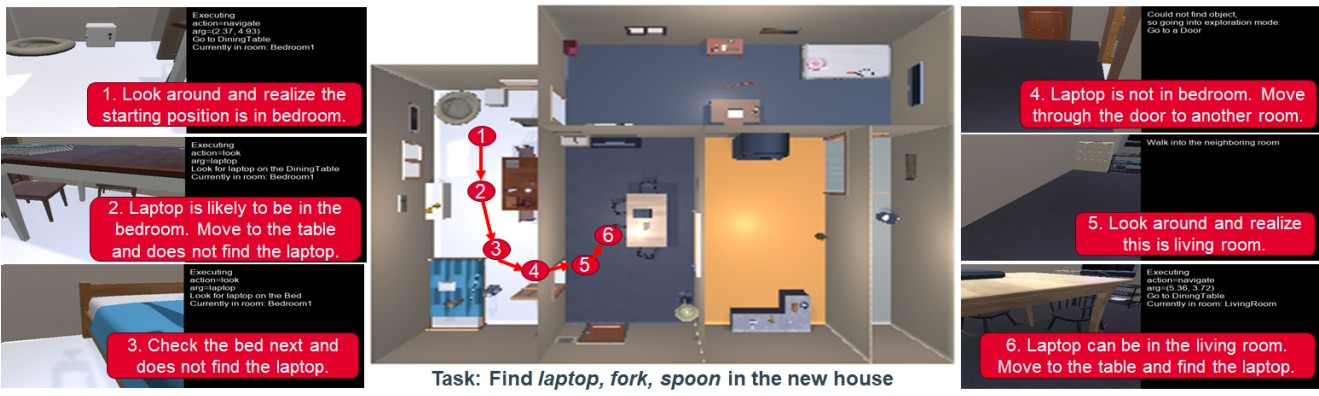

Figure 1: SayNav example: The robot uses LLM-based planner to efficiently find one target object (laptop) in a new house.

vironment.

The key innovation of SayNav is to incrementally build and expand a 3D scene graph of the new environment using perceived information during exploration. It then grounds feasible and contextually appropriate knowledge from LLMs which is used by the agent for navigation. This new grounding mechanism ensures that LLMs adhere to the physical constraints in the new environment, including the spatial layouts and geometric relationships among perceived entities. 3D scene graphs (Armeni et al. 2019; Kim et al. 2019; Rosinol et al. 2021; Hughes et al. 2022; Wald et al. 2020; Wu et al. 2021) have recently emerged as powerful high-level representations of 3D large-scale environments to support real-time decisions in robotics. A 3D scene graph is a layered graph which represents spatial concepts (nodes) at multiple levels of abstraction (such as objects, places, rooms, and buildings) with their relations (edges). We utilize this 3D scene graph representation to ground LLMs in the current environment, which is used to continuously build and refine the search plan for the agent during navigation.

Specifically, SayNav utilizes LLMs to generate step-by-step instructions on the fly, for locating target objects during navigation. To ensure feasible and effective planning in a dynamic manner, SayNav continuously extracts and converts a subgraph (from the current 3D scene graph) into a textual prompt to be fed to the LLMs. This extracted subgraph includes spatial concepts in the local region centered around the current position of the agent. The LLM then plans next steps based on this subgraph, such as inferring likely locations for the target object and prioritizing them. This plan also includes conditional statements and fallback options when any of the steps is unable to achieve the goal. For example, if the agent is not able to find the laptop on the desk, it will go to a next likely location (bed) in a bedroom.

SayNav also leverages LLMs to augment and refine the scene graph during navigation, such as annotating the room type based on current perceived objects. This improves the hierarchical organization of semantic information in the scene graph, that can support better planning. SayNav computes the feasibility of completing the current goal based on the room type which helps in better optimization of plan. For example, it can skip the restroom when looking for a spoon, but can come back later if needed.

SayNav only requires a few examples via in-context learning (Brown et al. 2020a; Ouyang et al. 2022) for configuring LLMs to conduct high-level dynamic planning to complicated multi-object navigation tasks in new environments. The LLM-generated plan is then executed by a pre-trained low-level planner that treats each planned step as a short-distance point-goal navigation sub-task (such as moving to perceived object A). This decomposition reduces the planning complexity of the navigation task, because the sub-tasks planned by LLMs are simple enough for low-level planners to execute successfully.

Figure 1 illustrates an example of SayNav utilizing LLMs to efficiently explore a new environment and locate one (laptop) of the three target objects. The agent first looks around (i.e., observes to build the scene graph) and identifies what type of room it starts from. After checking potential locations of the target objects in the room, the agent does not find any target. Then it decides to go through the door to move to another room. The agent continuously expands the scene graph during exploration and realizes that the neighbor room is a living room. There, it finds one target on the table and continues searching for other two objects.

The main contributions are summarized as follows.

1. We present, to the best of our knowledge, the first LLM-based high-level planner specifically for navigation tasks in large-scale unknown photo-realistic environments. The proposed LLM planner incrementally generates step-by-step instructions in a dynamic manner during navigation. The instructions generated from LLMs during navigation are consistent and non-redundant.

2. We propose a novel grounding mechanism to LLMs for navigation in new large-scale environments. SayNav incrementally builds and expands a 3D scene graph during exploration. Next-step plans are generated from LLMs, by utilizing text prompts based on a selected portion (subgraph) of the scene graph. Parts of the scene graph are also continuously refined and updated by LLMs.

3. We introduce a benchmark dataset for MultiON task across different houses for our evaluation and future use by researchers.

## 2 Related Work

In this section, we provide a brief review on related works in visual navigation, high-level planning with LLMs for autonomous agents and MultiON.

**Visual Navigation in New Environments** is a fundamental capability for many applications for autonomous agents. Recent learning-based approaches with DRL methods have shown great potential to outperform classical approaches based on SLAM (simultaneous localization and mapping) and path planning techniques, on different visual navigation tasks (Mishkin et al. 2019). These navigation tasks include point-goal navigation (Wijmans et al. 2019), image-goal navigation (Zhu et al. 2017), and object-goal navigation (Chaplot et al. 2020).

However, these methods generally require at least hundreds of millions of iterations (Wijmans et al. 2019) for training agents to generalize in new environments. This entails high cost in terms of both data collection and computation. In addition, it hinders the development of autonomous agents that can conduct more complex navigation tasks, such as multi-object navigation and cordon and search, that requires the ability to exploit common-sense knowledge and plan dynamically in novel environments.

Leveraging common-sense knowledge from LLMs allows us to avoid the high cost of training as in the previous learning-based methods. By effectively grounding LLMs (such as ChatGPT) via text prompting, our approach enables efficient high-level planning for visual navigation in unknown environments. To better demonstrate and evaluate our proposed method, we use multi-object navigation, which is more complex than previous navigation tasks such as object-goal navigation. The multi-object navigation task demands common-sense knowledge, as humans do, to efficiently search for multiple different objects in large-scale unknown environments.

**High-Level Planning with LLMs** has become an emergent trend in the robotics field. LLMs by virtue of both training on internet scale data and instruction tuning have demonstrated excellent capabilities to perform zero/few shot learning for unseen tasks (Zhao et al. 2023; Brown et al. 2020b). Recent instruction tuned models such as ChatGPT have further shown strong capabilities to follow natural instructions expressed as prompts (Chung et al. 2022; Peng et al. 2023).

Recent works in autonomy have used LLMs and demonstrated significant progress (Ahn et al. 2022; Song et al. 2022; Huang et al. 2022; Liu et al. 2023; Driess et al. 2023; Brown et al. 2020a; Ouyang et al. 2022) in incorporating human knowledge, that enables efficient training of autonomous agents for tasks such as mobile manipulation. These works reduce the learning complexity by using a two-level planning architecture. For each assigned task, they utilize LLMs to generate a high-level step-by-step plan. Each planned step, formulated as a much simpler sub-task, can be executed by an oracle (ground truth) or a pre-trained low-level planner that maps one step into a sequence of primitive actions. Agents with these LLM-based planners are able to perform a new task with only a few training examples via in-context learning (Brown et al. 2020a; Ouyang et al. 2022).

However, these LLM-based planners have two major limitations for visual navigation tasks in new large-scale environments. First, the grounding mechanisms in these methods (Ahn et al. 2022; Song et al. 2022; Huang et al. 2022; Liu et al. 2023) are designed for small-scale environments. For example, works such as (Song et al. 2022; Singh et al. 2023) have focused on the AI-THOR based environment that consists of only a single room. Moreover, these methods only rely on detection of specific objects. They do not consider room layout and the topological arrangement of perceived entities inside the room, which are important to ground LLMs in the physical environment for visual navigation tasks. Therefore, knowledge extracted from LLMs using these methods might not be contextually appropriate to an agent for navigation in large-scale settings, such as multi-room houses.

Second, some of these LLM-based planners typically generate a multi-step long-horizon plan in the beginning for the assigned task, which is not feasible for navigating in unknown environments. They also lack the capability to change the plan during task execution. In contrast, an effective search plan for navigation in new places is required to be incrementally generated and updated during exploration. Future actions are decided based on current perceived scenes with the memory of previously-visited regions.

SayPlan (Rana et al. 2023) addresses the first issue by using a pre-built ground-truth 3D scene graph of a known large-scale place, to ground LLMs for high-level task planning. However, the planning complexity for SayPlan is simplified due to the availability of the entire ground-truth scene graph prior to task execution. In other words, SayPlan cannot be used for task planning in unknown environments.

Our approach, SayNav, is designed to leverage LLMs specifically for visual navigation in unknown large-scale environments. We propose a new grounding mechanism that incrementally builds a 3D scene graph of the explored environment as inputs to LLMs, for generating the high-level plans. SayNav also dynamically generates step-by-step instructions during navigation. It continuously refines future steps based on newly perceived information via LLMs.

The only work we found to leverage LLMs specifically for navigation tasks in unknown environments is L3MVN (Yu et al. 2023). It uses LLMs to find the nearby semantic frontier based on detected objects, for expanding the exploration area to eventually find the target object. For example, moving to the (sofa, table) region which is more likely to have TV. In other words, it utilizes LLMs to hint to the next exploration direction. It does not use LLMs as a full high-level planner, that generates step-by-step instructions. In contrast, our SayNav uses the 3D scene graph to ground LLMs as a high-level planner. Our LLM-based planner generates the instructions in a dynamic manner, and considers its prior planned steps to generate better future plans.

**MultiON** task has attracted attention of researchers in the last few years. As already mentioned, most of them (Chen et al. 2022; Zeng et al. 2023; Marza et al. 2022, 2023) have considered pre-defined sequence of objects to be localized which simplifies the task by a great extent. Moreover, they employ pure DRL-based approaches and hence suffer from

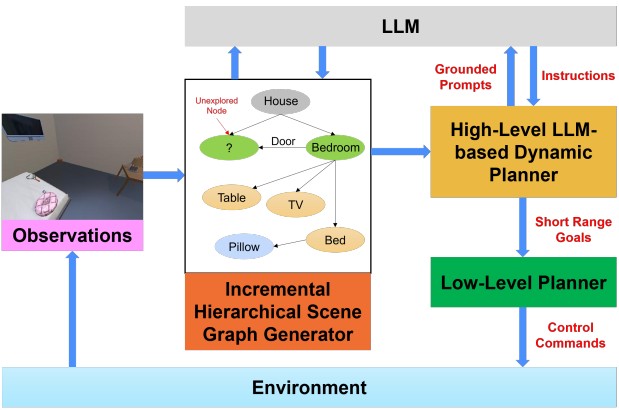

Figure 2: The overview of our SayNav framework.

issues discussed before. (Gireesh et al. 2023) is the only work to consider MultiON without any sequence of objects but again makes use of a DRL-based approach.

# 3 SayNav

We now describe SayNav's framework as well as the multi-object navigation task.

## 3.1 Task Definition

We choose Multi-Object Navigation task, to validate Say-Nav. The goal of this task is to navigate the agent in a large-scale unknown environment in order to find an instance for each of three predefined object categories (such as "laptop", "tomato", and "bread"). The agent is initialized at a random location in the environment and receives the goal object categories $(o_i, o_j, o_k)$ as input. At each time step $t$ during navigation, the agent receives environment observations $e_t$ and takes control actions $a_t$. The observations include RGBD images, semantic segmentation maps, and the agent's pose (location and orientation). The action space includes five control commands: $turn\text{-}left$, $turn\text{-}right$, $move\text{-}forward$, $stop$, and $look\text{-}around$. Both $turn\text{-}left$ and $turn\text{-}right$ actions rotate the agent by 90 degrees. The $move\text{-}forward$ action moves the agent by 25 centimeters. The task execution is successful if the agent locates (by detection) all three objects within a time period.

Note that multi-object navigation task poses much larger planning challenges and task complexities than previous navigation tasks, which either look for a single object (Chaplot et al. 2020) or reach a single target point (Wijmans et al. 2019). For example, the agent needs to dynamically set up the search plan based on the prioritized order among three different objects. This plan can also be changed during exploration in a new house with unknown layouts. As shown in Figure 1, the agent first realizes that it is in the bedroom and then decides to prioritize places (such as the table) to locate the laptop inside this room. On the other hand, if the agent had started in the kitchen, it would have been more efficient to search for the fork and spoon first. Therefore, this new task requires extensive semantic reasoning and dynamic planning capabilities, as what humans possess, for an

autonomous agent to explore in large-scale unknown environments.

## 3.2 Overview

SayNav's framework is illustrated in Figure 2. The corresponding pseudo-code is in Algorithm 1. It includes three modules: (1) Incremental Scene Graph Generation, (2) High-Level LLM-based Dynamic Planner, and (3) Low-Level Planner. The Incremental Scene Graph Generation module accumulates observations received by the agent to build and expand a scene graph, which encodes semantic entities (such as objects and furniture) from the areas the agent has explored. The High-Level LLM-based Dynamic Planner continuously converts relevant information from the scene graph into text prompts to a pre-trained LLM, for dynamically generating short-term high-level plans. Each LLM-planned step is executed by the Low-Level Planner to generate a series of control commands for execution.

## 3.3 Incremental Scene Graph Generation

This module continuously builds and expands a 3D scene graph of the environment being explored. A 3D scene graph is a layered graph which represents spatial concepts (nodes) at multiple levels of abstraction with their relations (edges). This representation has recently emerged as a powerful high-level abstraction for 3D large-scale environments in robotics. Here we define four levels in the 3D scene graph: small objects, large objects, rooms, and house. Each object node is associated with its 3D coordinate and room node is associated with its bounds. Every door is treated as an edge between two rooms, which also has an associated 3D coordinate. All other edges reveal the topological relationships among semantic concepts across different levels. Figure 3 shows one example of our scene graph. Mathematically, our scene graph can be represented as a set of 4 kinds of triplets:

$$\{(s_h, `near`, l_i), (l_i, `in`, r_j), (r_j, d_{jk}, r_k), (r_j, `in`, H)\}$$

where, $s_h$: small object, $l_i$: large object, $r_j, r_k$: rooms ($i \neq j$), $d_{jk}$: door between $r_j$ & $r_k$, $H$: house (root node).

The scene graph is built using environmental observations received by the agent during exploration. The depth of each segmented object can be obtained based on RGBD images and semantic segmentation images. The 3D coordinate of each perceived object can then be estimated by combining its depth information at multiple timestamps and the corresponding agent's poses.

We also utilize LLMs to augment and refine high-level abstractions of the scene graph. For example, we use LLMs to annotate and identify the spatial entity (room type) at the room level of the graph based on its connected objects at lower levels. For instance, a room is probably a bedroom if it includes a bed. The bounds of a room are calculated based on the detection of walls and floor of the room.

SayNav also uses the 3D scene graph to support memory for future planning. For example, it automatically annotates the rooms that have been investigated. Therefore, it will not generate repeated plans when the agent revisits the same room during exploration.

**Input** : Start location of robot *start_location*
house ID *house_id*
Target Objects *target_objects*

1 *unfound_objects* ← *target_objects*
2 spawn_robot(*start_location*)
3 *SceneGraph* ←
   create_scene_graph(*house_id*)
4 **while** *len(unfound_objects)* > 0 **do**
5    *objs_found, observations* ←
     look_around()
6    update_unfound_objects(*objs_found*)
7    *room_type* ←
     identify_room_type(*observations*)
8    *SceneGraph*.update(*room_type, observations*)
9    *plan_needed* ← is_feasible(*room_type*)
10    **if** *plan_needed* **then**
11       *subgraph* ← *SceneGraph*.
12       extract_subgraph(*room_type*)
13       *plan* ←
        query_llm_for_plan(*subgraph,*
        *unfound_objects*)
14       **for** *action* **in** *plan* **do**
15          **if** *action.type* = *'navigate'* **then**
16             navigate_to(*action.target_location*)
17          **end**
18          **else if** *action.type* = *'look'* **then**
19             *objs_found, observations* ←
             look_around()
20             *SceneGraph*.update(*observations*)
21             update_unfound_objects(*objs_found*)
22          **end**
23       **end**
24    **end**
25    **if** *len(unfound_objects)* > 0 **then**
26       **if** *SceneGraph.all_doors_explored()* **then**
27          *return* 'Task Failed'
28       **end**
29       *door* ←
        find_next_unexplored_door()
        navigate_to(*door*)
30    **end**
31 **end**
32 *return* 'Success'

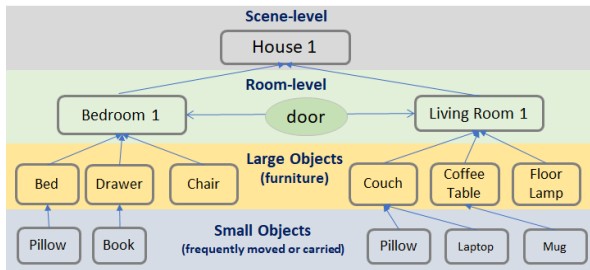

Figure 3: An example of our scene graph.

Our high-level LLM-based dynamic planner extracts a subgraph from the full 3D scene graph and converts it into text prompts, which are fed to an LLM. The extracted subgraph includes spatial concepts in the local region centered around the current position of the agent. We implemented the LLM prompts similar to (Singh et al. 2023), which constructs programming language prompts based on the text labels in the extracted subgraph. Once prompts are received, the LLM planner outputs short-term step-by-step instructions, as pseudo code. The generated plan provides an efficient search strategy within the current perceived area based on human knowledge, prioritizing locations to visit inside the room based on the likelihoods of target objects being discovered. For example, LLM may provide a plan to first check the desk and then the bed to find the laptop in the bedroom. Figure 4 shows the prompt structure used to generate the plan. We provide two in-context examples inside the prompt to constrain the LLM-generated plans. For instance, we constrain each step to generate a *navigate* or *look* function call with arguments and a high-level comment.

The LLM-based planner also extends and updates the plan when the previous plan fails or the task goal (finding three objects) is not achieved after the previous short-term plan executes.

### 3.5 Low-Level Planner

The Low-Level Planner converts each LLM-planned step into a series of control commands for execution. To integrate two planners, SayNav formulates each LLM-planned step as a short-distance point-goal navigation (POINTNAV) sub-task for the low-level planner. The target point for each sub-task, such as moving from the current position to the table in the current room, is assigned by the 3D coordinate of the object (e.g. table) described in each planned step.

SayNav's low-level planner takes the RGBD images (resolution $320 \times 240$) and the agent's pose (location and orientation) as inputs, and it outputs move_forward, turn_left and turn_right actions to control the robot following standard POINTNAV settings. Note that large-scale DRL approaches typically take $10^8$ to $10^9$ simulation steps during training to solve POINTNAV tasks in simulation environments (Wijmans et al. 2019; Weihs et al. 2020), which poses serious training requirements and computation demands. In SayNav, however, the two-level planning architecture simplifies the job of the low-level planner. The low-level planner mostly outputs control commands for

### 3.4 High-Level LLM-based Dynamic Planner

Similar to previous works in LLM-based planning, SayNav utilizes a two-level planning architecture to reduce the learning complexity of the assigned task. However, instead of generating a complete high-level plan for the entire task in the beginning, SayNav utilizes LLMs to incrementally generate a short-term plan regularly, based on current observations and the memory of previously-visited regions. This high-level planner can be set-up using only a few training examples via typical in-context learning procedures (Brown et al. 2020a; Ouyang et al. 2022) (as shown in Figure 4).

**Search Plan prompt**

**System**
Assume you are provided with a text-based description of a room in a house with objects and their 2D coordinates

Task: I am at a base location. Suggest me a step-wise high-level plan to achieve the goal below. Here are some rules:
1. I can only perform the actions- (navigate, (x, y)), (look, objects)
2. I can only localize objects around other objects and not room e.g. apple should be looked for on the table and not kitchen.
3. Provide the plan in a csv format. I have shown two examples below:
a) Goal = 'Find a laptop', Base Location = 'LivingRoom'
- navigate; (3.4, 2.6); Go to table
- look; (laptop); Look for laptop near the table
b) Goal = 'Find apple and bowl', Base Location = 'Kitchen'
- navigate; (3.4, 2.6); Go to DiningTable
- look; (apple, bowl); Look for apple and bowl near the DiningTable
- navigate;  (8.32, 0.63); Go to CounterTop
- look; (apple, bowl); Look for apple and bowl near the CounterTop
4. Start the line with - and no numbers
5. Provide the plan as human would by e.g. based on contextual relationships between objects

**User**
Room description
{room_graph}

Goal: {goal}
Base location: {base_location}

Figure 4: Prompt used to create the search plan for a particular room

short-range movements. The LLM-based high-level planner is also robust to failures in the low-level planner, by making regular plan updates. In this way, the training load required on the low-level planner can be greatly reduced.

Encouraged by the success of imitation learning (IL) on navigation tasks under resource constraints (Ramrakhya et al. 2022, 2023; Shah et al. 2023), we investigate a sample efficient IL-based method to train the low-level planner for the agent in SayNav. This low-level planner is trained from scratch (without pretraining) on only 2800 episodes or $7 \times 10^5$ simulation steps. Specifically, the low-level planner is trained using the DAGGER algorithm (Ross et al. 2011) to follow a shortest path oracle as the expert. Despite the fact that the shortest path oracle lacks the exploration behavior required to solve the POINTNAV task in complex environments (*e.g.* multiple rooms), we find that it helps the agent to learn short-distance navigation skills very quickly, without human-in-the-loop. More details about our low-level planner are available in the supplementary material.

## 4 Experimental Results

**Dataset:** Most prior Embodied AI simulators such as AI2-THOR (Kolve et al. 2017) or Habitat (Szot et al. 2021) are either based on environments with single rooms or lack the natural placement of objects within the environment or lack the ability to interact with the objects. For our experiments, we opted for the recently introduced ProcTHOR framework (Deitke et al. 2022), which is built on top of the AI2-THOR simulator. ProcTHOR is capable of procedurally generating full floor plans of a house given a room specification (ex: a house with 2 bedrooms, 1 kitchen and 1 bathroom). It also populates each floorplan with 108 object types, with realistic, physically plausible, and natural placements. The scenes in ProcTHOR are interactive which allows to change the state, lighting and texture of objects, posing a bigger chal-

lenge for perception and a broader scope for future work. We build a dataset of 132 houses with 3-10 rooms each and select three objects for each house to conduct multi-object navigation task. More details about our dataset are available in the supplementary material.

**Metrics:** We report two standard metrics that are used for evaluating navigation tasks: Success Rate (SR) and Success Weighted by Path Length (SPL) (Anderson et al. 2018). SR measures the percentage of cases where the agent is able to find all the three objects successfully, while SPL normalizes the success by ratio of the shortest path to actual path taken. We use the minimum of the shortest path from the starting point to permutations of all the target objects. In addition to these two metrics, we measure the similarity between the object ordering obtained by the agent and that by the ground-truth. The ground-truth object ordering gives an idea of how a perfect agent would have explored the space by first identifying objects that are highly probably in current room/scene-graph and then exploring other rooms. We use the Kendall distance metric (Lapata 2006), which computes the distance between two rankings based on the number of disagreeing pairs. We use the Kendall Tau that converts this distance into a correlation coefficient, and report it over the successful episodes (all three targets are located).

**Implementation Details:** We use the default robot with head-mounted RGBD camera in the AI2-Thor simulator. The camera has $320 \times 240$ resolution with 90°field-of-view (FoV). The details of the robot observations and actions can be referred to the section 3.1.

We conduct experiments with 2 different LLMs: *gpt-3.5-turbo* and *gpt-4*. For training the low-level planner, we used the IL method, described in the section 3.5. It achieves $84.5\%$ POINTNAV success rate (success radius 1.5m, max 300 steps) and $0.782$ SPL in unseen ProcThor-10k-val scenes with random start and goal locations. For short-range movements within a single room, performance increases to $98.5\%$ success rate and $0.930$ SPL. More details are available in the supplementary material.

Note that SayNav consists of three modules– incremental scene graph generator, LLM-based planner, and a low-level planner. The major goal of our experiments is to fully validate and verify the LLM-based planning capabilities in SayNav for MultiON. Therefore, for each of the other two modules, we implemented an alternative option which uses ground truth information to avoid any error within that module. This allows us to conduct ablation study for determining the impact of each module on the overall performance.

First, we allow the scene graph to be generated using ground truth (**GT**) instead of visual observations (**VO**). We have described the generation of scene graph using VO in section 3.3. The GT option directly uses the ground truth information of surrounding objects, including 3D coordinates and geometric relationships among objects, to incrementally build the scene graph during exploration. This option avoids any association and computation ambiguity from processing on visual observations, such as computing 3D coordinates for each object based on RGBD image and its segmentation.

Second, we use an oracle planner (**OrNav**) as the low-level planner instead of our efficiently-trained IL agent

| | Scene Graph | LL Planner | SR (%) | SPL | Kendall Tau |
|---|---|---|---|---|---|
| Baseline | | | 56.06 | 0.49 | |
| SayNav (gpt-3.5) | GT | OrNav | 95.35 | 0.43 | 0.70 |
| | GT | PNav | 80.62 | 0.32 | 0.72 |
| | VO | OrNav | 71.32 | 0.48 | 0.56 |
| | **VO** | **PNav** | **60.32** | **0.34** | **0.62** |
| SayNav (gpt-4) | GT | OrNav | 93.93 | 0.46 | 0.76 |
| | GT | PNav | 84.09 | 0.36 | 0.78 |
| | VO | OrNav | 69.80 | 0.47 | 0.76 |
| | **VO** | **PNav** | **64.34** | **0.33** | **0.78** |

Table 1: Results of SayNav on multi-object navigation task. **Baseline** uses a PNav agent to navigate along the shortest route among targets based on ground-truth positions; **GT** and **VO** build the scene graph from ground-truth object/room locations and visual observations provided by the simulator respectively; **OrNav** and **PNav** use oracle and IL-learned low-level (**LL**) planner respectively for navigating between the points assigned by the high-level planner.

(**PNav**). We have described the implementation of PNav in the section 3.5. For OrNav, we use an A* planner which has access to the map of the environment. Given a target location, it can plan the shortest path from the agent's current location to the target.

**Baseline:** (Gireesh et al. 2023) didn't release their source code or the dataset used by them, so it is not straightforward to compare our approach with them. Hence, we implemented a strong baseline method that could potentially reflect the upper bound of the performance of a learning-based agent using the same amount of training data as ours. The baseline agent uses two privileged information that SayNav doesn't have access to. First, it has access to the optimal order of target objects which achieves the shortest path. This simplifies the MultiON task to become a series of ObjectNav tasks. Second, as a PointNav policy performs better than an ObjectNav policy, we also provide the baseline agent access to the ground truth coordinate of each target object, which then simplifies the task to a series of PointNav tasks. In other words, we implement a PointNav agent to navigate through ground truth points of the objects in the optimal order. However, even with a reasonable PNav agent (98.5% SR for short-distance navigation), SR decreases substantially for our task. It is because of the difficulty in successful execution of multiple (sequential) point-goal navigation sub-tasks, including cross-room movements.

### 4.1 Quantitative Results

Table 1 shows the results of the baseline along with different implementation choices in SayNav. Note for the baseline method, even after using ground-truth object locations in optimal order, SR is only 56.06%, which indicates the difficulty of multi-object navigation task. In comparison, SayNav, without using any ground truth, achieves a higher SR (60.32% and 64.34% with gpt-3.5-turbo and gpt-4 respectively). This improvement highlights the superiority of Say-

Nav in navigating in large-scale unknown environments.

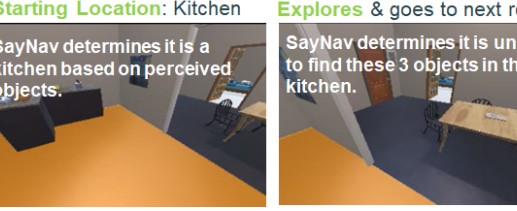

**Goal: Find *alarmclock, laptop, cellphone***

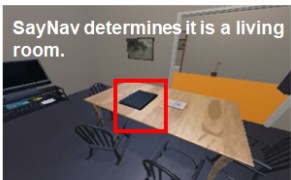
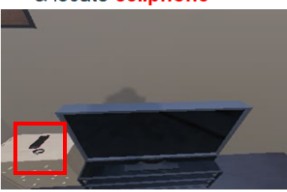

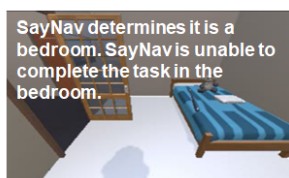
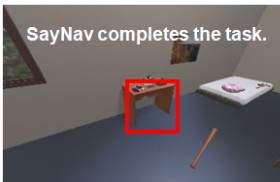

Figure 5: Visualization of an episode with SayNav (OrNav + GT) for multi-object object navigation task.

**SR:** With SayNav, we observe that the best performance is achieved when using scene graph generated by ground-truth object/room location from the simulator (**GT**) along with **OrNav**. When we replaced **GT** with **VO**, we do observe a loss in performance. We found that the drop in SR can be associated with various challenges encountered in any perception based algorithms. The inaccurate estimation of objects' 3D positions due to partial observations can lead to failures in detecting targets and navigation. In addition, we remove objects less than 20 pixels on semantic segmentation observations for more practical behavior. Therefore, very small objects can also be missed-out while building the scene graph from visual observations. We also observed a specific challenge in **VO** associated with estimating the location of glass doors. From the depth map, the depths of visible objects behind the glass door represent the depths of the actual physical door, which fails the estimation of the location of the door. A similar trend can be found from results with **GT** & **PNav** and with **VO** & **PNav**. When replacing **OrNav** with **PNav**, we also observe a fall in performance. This is obvious as **PNav** doesn't access any ground truth information, as compared to **OrNav**.

However, even with all these challenges, SayNav outperforms the oracle-based baseline and succeeds in multi-object navigation tasks. We believe it is due to LLM-based dynamic planning capabilities, with the grounding mechanism based on incremental scene graph generation. It leverages common-sense knowledge, as humans do, to efficiently

search multiple different objects in an unknown environment. It also refines or corrects the plan in case of any failure in a planned step.

**SPL:** Looking at the SPL metric, we see a drop in Say-Nav as compared to the baseline. Note that SPL reflects the length of the path taken by the agent as compared to the shortest possible path. For example, SPL=1 for an episode would mean that the agent, starting from initial position, goes straight to the targets along the shortest path in the optimal order with zero exploration which is practically impossible in an unknown environment. As a result of access to the ground-truth object locations in optimal order, it becomes obvious for the baseline to have higher SPL. From the results, we also observe that the low-level planner has the major impact on SPL. The system achieves higher SPL with **OrNav** as compared to **PNav**.

**Kendall-Tau:** The Kendall-Tau ($\tau$) metric measures the similarity between the order of objects as located by the agent and the optimal ordering based on the ground-truth. It shows the importance of the knowledge provided by the LLMs, for finding the optimal plan. We observe that the ordering of the objects is not affected much by the low-level planner. This is reasonable since the ordering should majorly depend on the plans generated by the high-level planner. As expected, the score drops when we replace **GT** with **VO** since LLM uses scene-graph to generate the plan. The considerable improvement in the score with gpt-4 (vs gtp-3.5-turbo) shows that using a better LLM enables an improvement in use of common-sense priors and yields more optimal ordering. Also, note that Kendall Tau metric doesn't apply to the baseline since it already has access to the optimal order of targets.

### 4.2 Qualitative Results

We show an example of a typical episode in Figure 5 where the agent is asked to locate an alarm-clock, a laptop, and a cellphone in an unknown house. The agent happens to start in the kitchen (determined by LLM based on perceived objects). The planner reasons that it is unlikely to find either of the objects there, so it decides to go to another room through a door. Then, it comes to a living-room where it is able to locate the laptop and cellphone. The third object still remains unfound, so it again decides to go to another room via a door. Eventually, it locates the alarm-clock in the final room. The complete demo-video for this example can be found in the supplementary material.

### 5 Limitations and Future Work

This section discusses the limitations of our work and some ideas to improve it in future. As mentioned before, our scene graph generation faces various challenges encountered in any perception-based algorithms. In addition to the glass door issue that we described earlier, Figure 6 shows a failure case for another issue due to visual observations. Note, in our experiments, the agent is not equipped with arms to open/close the door. Therefore, it only can go through open doors to move to other rooms. In this episode, the agent from most of the positions in the room cannot observe that the

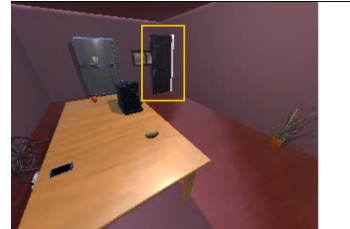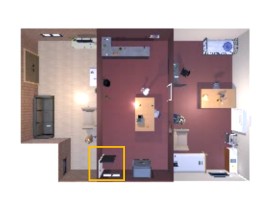

Figure 6: A failure example: The left picture shows the RGB image from the camera mounted on the agent and the right picture shows the top-down view of the house. Due to geometry of the room, agent is unable to observe that the door is open and hence, unable to navigate through the door (marked with yellow rectangle).

door is open (which connects to the other room that has the target object). The robot repeatedly tries to go towards the center of the current room and refine the scene graph. However, it is still not able to identify the "open" status for the door, and therefore, it fails to achieve the goal in the end.

A better mechanism to verify attributes (open/close) associated with the object node (door) in the scene graph can help to alleviate this case. For example, the agent can move closer to the door, verify visual observations from all possible angles, and compare the depth observation from the door to depth information from the wall (closed doors shall have nearly identical depths as the connected wall).

In the future, we also plan to evaluate SayNav with an actual semantic segmentation network such as SAM (Ji et al. 2023). This will further study the impact of noise in the perception sensor and the robustness of SayNav. We would also like to deploy SayNav on a real robot for experiments. It will help to validate the generalization capability of Say-Nav from simulation to the real world. It will also be interesting to explore the possibility of using an open-source instruction-tuned LLM, such as Vicuna (Peng et al. 2023) instead of GPT-4 and GPT-3.5 in SayNav. We believe that it may generate more contextually-suitable plans for SayNav using these custom tuned LLMs.

### 6 Conclusion

We present SayNav, a new approach for efficient generalization to complex navigation tasks in unknown large-scale environments. SayNav incrementally builds and converts a 3D scene graph of the explored environment to LLMs, for generating dynamic and contextually appropriate high-level plans for navigation. We evaluate SayNav on the multi-object navigation task, that requires the agent to efficiently search multiple different objects in an unknown environment. Our results demonstrate that SayNav even outperforms a strong oracle based Point-nav baseline (64.34% vs 56.06%), for successfully locating objects in large-scale new environments. The benchmark dataset and the source code will be released upon the acceptance of the article.

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
