# OpenReview forum: "SayNav: Grounding Large Language Models for Dynamic Planning to Navigation in New Environments"
_icaps-conference.org/ICAPS/2024/Conference — ICAPS 2024_

### Official Review · Reviewer_ZGtQ · 2024-01-13

**Significance And Importance:** 1
**Soundness:** 2
**Novelty:** 2
**Clarity:** 2
**Overall Evaluation:** 1
**Confidence:** 3

**Weaknesses:**

0: Minor weaknesses requiring some work to be addressed for the paper to be accepted.

**Contributions Of The Paper:**

The paper introduces SayNav, an autonomous navigation approach using semantic reasoning and dynamic planning. Leveraging human knowledge from Large Language Models (LLMs), SayNav builds a 3D scene graph for efficient generalization in unknown environments. It generates high-level navigation plans with LLMs and executes them using a pre-trained low-level planner. The system dynamically generates step-by-step instructions and refines them based on real-time information

**Ethical Considerations:**

(1) Not Applicable: The paper does not have any ethical considerations to address

**Nomination For Best Paper:**

No

**Questions For Authors:**

In Section 3.3, the statement "Every door is treated as an edge between two rooms" raises a question: How is an edge computed if there is no door between two rooms or if a door is not detected by the system?

What are the roles of small objects and large objects and how they are defined?

How to determine whether the current region has been explored or not? Consider a scenario where there are two distinct pathways from a living room to a room. If an agent enters the room through one pathway and returns to the living room via the other, how can we ascertain that it's the same region explored previously? Do the authors maintain pose information for explored objects and regions as a means of recordkeeping?

Questions arise regarding the validation of short-term plans obtained through LLM: Is there any logic-based reasoning employed for this purpose to validate the plan?

Additionally, the inquiry extends to feedback mechanisms when faced with failures in LLM-generated plans: What type of feedback is provided in such instances?

Another consideration is how the approach addresses knowledge uncertainty within LLM.

**Reproducibility:**

3: Authors describe the implementation and domains in sufficient detail.

**Strengths Of The Paper:**

The paper addresses a compelling exploration problem and employs experiments to evaluate the proposed approach. Nevertheless, there are significant issues related to the approach that require attention and clarification from the authors.

**Weaknesses Of The Paper:**

Certain aspects of the approach, particularly concerning graph construction and the distinction between similar and identical objects, lack clarity.

To validate the proposed approach, the authors should conduct real experiments.

Moreover, there is a notable absence of a reasoning process to verify the correctness of the high-level plans generated by Large Language Models (LLMs). Given that LLMs may occasionally fail to produce accurate plans, the inclusion of a verification mechanism is essential.

Several other papers extract knowledge from Large Language Models (LLMs), such as "Language Models as Zero-Shot Planners: Extracting Actionable Knowledge for Embodied Agents." It is crucial for the authors to compare their approach with these existing methods.

---

> ### Author Rebuttal · Authors · 2024-01-28
>
> We thank the reviewer for the valuable feedback.
>
> **Edge:** There need not exist an edge between every pair of room nodes. If there is no door between two rooms, then there is no edge between their nodes.
>
> **Large vs Small Objects:** Large objects act as landmarks which are visible from far. LLM uses these large objects to reason if an object can be found near them. For ex, it may be logical to walk towards *dining table* to find a *knife*. The decision of small vs. large object is made based on its dimensions and LLM’s understanding about the object class.
>
> **RecordKeeping:** Once a room has been explored, the high-level planner updates the metadata of the room node.  We have also shown an alternative way of supporting memory in **Appendix (Sec C)** where the LLM itself keeps track of which rooms have been explored. We also store the mean positions of the objects and rooms in the scene graph which allows the agent to recognize if the current room has been previously explored.
>
> **Existing Methods:** Most existing approaches including the work pointed by the reviewer operate in known and/or small-scale environments. The mentioned work assumes the oracle low-level actions and generates a plan for the whole task at once. On contrary, we generate the plans on the fly during exploration of environment and keep updating the plan as new information comes in.
>
> **Validation & Feedback:** Regarding the validation of LLM plans, we would like to point to the results with oracle planner and ground-truth based scene graph, shown in **Table 1**. In this setting, the only module which isn’t perfect is the LLM-based high-level planner. Success Rate of 95.35% in this setting reflects the strength of our LLM-based planner. We agree that there can be errors associated with LLM outputs (e.g. hallucinations). One technique that we employ to improve reliability is making LLM output a comment for every step in the generated plan. In case of failures, LLM is informed that the plan has failed and is asked to generate a new plan. Having said that, we agree that we can have more verification and feedback mechanisms in future work to further improve the performance.
>
> **Real Experiments:** We have successfully ported SayNav to a real robot, with test analysis and demo videos available. However, robot integration was beyond the focus of this paper, so we didn’t include them. We are unable to provide video links to demos here due to conference rules but will add them in the camera ready.

---

### Official Review · Reviewer_tcTR · 2024-01-22

**Significance And Importance:** 2
**Soundness:** 3
**Novelty:** 3
**Clarity:** 3
**Confidence:** 3

**Weaknesses:**

1: Minor weaknesses that are easily fixable.

**Contributions Of The Paper:**

The paper looks at the use of LLMs as a high-level planner for effectively performing multi-objective navigation tasks. The idea being that the planner could leverage human common-sense knowledge encoded in LLMs to improve planning in such scenarios. The other major components of the system involve a mechanism to build a scene graph, which is primarily used to create the prompts, and a low-level planner to drive the actual navigation. The method is tested on a dataset generated from ProcTHOR against a baseline that has access to some ground truth information. The initial results seem promising.

**Ethical Considerations:**

(1) Not Applicable: The paper does not have any ethical considerations to address

**Nomination For Best Paper:**

No

**Overall Evaluation:**

-1: (weak reject)

**Questions For Authors:**

I would like to ask a question related to each of the three points I raised above.

1. Can the proposed method handle non-ergodic domains?

2. How does the method handle the failure of the low-level planner?

3. How can the proposed method beat the baseline with access to the ground truth?

**Reproducibility:**

3: Authors describe the implementation and domains in sufficient detail.

**Strengths Of The Paper:**

There is a lot of interest and skepticism regarding using LLMs for planning tasks. The paper here presents a very intuitive use of LLMs where one can see the strengths of LLMs (such as a repository of everyday knowledge) used to improve the planning process.

**Weaknesses Of The Paper:**

1. Task being solved - The paper frames itself as introducing a way to leverage LLMs to perform “dynamic planning” in “novel environments.” These terms never get a very clear definition, but I am interpreting this to mean online planning in partial observability.  MultiOn is presented as a demonstration of this capability. I want to push back a little bit on that. As far as I can see, the method seems pretty ill-equipped to handle any non-ergodic settings, which could be a very important factor in acting in any environment with some level of uncertainty. I would highly encourage the authors to flip the contribution with a focus on MultiOn, with potential implications for more general problems.

2. Refinements provided by low-level planner - One thing I was unclear about was what happens when the low-level planner fails to provide any kind of refinement. This could happen for multiple reasons, say the high-level plan may be impossible to refine correctly or maybe due to incompleteness or time constraints placed on the low-level planner. I didn’t see any clear discussion on how the high-level planner handles such scenarios. There is a line in section 3.4 about incorporating plan failures, but I wasn’t sure if it was referring to low-level plan failures and, if so, what kind of information is included. Also, if the 3D object coordinates are available, why is a visual input needed at all? Can’t you use just traditional motion planning algorithms here?

3. Baseline - To be honest, this would have been a good opportunity to compare the method to a purely traditional planning approach. One potential candidate the authors could have considered would have been some variation of the method presented in [1]. Regardless, I am still unsure about why the proposed method outperforms the baseline (it is possible that I am misunderstanding the baseline). Both the proposed method and the baseline use the same low-level planner, and the baseline has access to both the optimal order in which the objects need to be visited and where the objects are located. One of the strengths the authors highlight is that the LLM-based planner would easily recognize what each room is from observing a few objects, but I imagine this can’t compare to a baseline that knows where the objects are.

[1] Liu, Xiaotian, Hector Palacios, and Christian Muise. "Egocentric Planning for Scalable Embodied Task Achievement." arXiv e-prints (2023): arXiv-2306.

---

> ### Author Rebuttal · Authors · 2024-01-28
>
> We thank the reviewer for the insightful comments.
>
> **Non-ergodic:** SayNav uses commonsense knowledge from LLM to generalize planning across environments. It generates and updates plans on the fly during exploration and our incremental scene graph keeps refining itself using latest observations. This enables SayNav to handle some levels of uncertainty in the environment. We’ve also ported SayNav to a real robot, with tests in a cafeteria with dynamic people. We will include it in camera ready.
>
> **Handling failures of Low-Level Planner (LLP):** LLP can fail in 2 ways:
>
> a. Unable to locate the objects based on the plan created by High-Level Planner (HLP).
>
> b. Fails to execute the plan/gets stuck (ex: unable to cross the door, which is common in RL- based PointNav agents).
>
> HLP avoids **Failure-a** by generating a plan that considers alternate locations (prioritized by semantics) for the object. However, if the goal is not achieved then HLP puts LLP into an exploratory mode (ex: try to go to next room). HLP then generates a new plan once it receives new information about the environment. For **Failure-b**, the HLP detects this situation based on the current location of the agent. In this case, it attempts to refine the scene graph (ex: refine door’s position) by instructing the LLP to randomly move around and collect more observations.
>
> **Baseline:** The work pointed out by the reviewer assumes fully observable environments and hence, is not applicable to our problem setting. The traditional motion planning approaches for unknown environments require dense mapping via SLAM based algorithms. Many previous works (ex: Savva M. et al. 2019, arxiv:1904.01201) have shown that RL / IL based Point-goal Navigation (PointNav) outperforms SLAM-based approaches in unknown environments by a large extent (80% vs 62% in the mentioned example). So, we chose to implement the strongest possible baseline agent which employs IL-based PointNav policy (see **L530-550**). However, one drawback with PointNav policy is its loss in performance in large-scale environments. As shown in **L500-508**, the PointNav policy performs better on short-range goals (98.5% SR, 0.93 SPL) than long-range goals (84.5% SR and 0.782 SPL). Although, the baseline has access to the locations of the goal objects, finding each object still requires it to plan a long-range path across multiple rooms. SayNav solves this issue by generating only short-term goals, which are handled more efficiently by LLP.

---

### Official Review · Reviewer_ssf1 · 2024-01-23

**Significance And Importance:** 3
**Soundness:** 3
**Novelty:** 4
**Clarity:** 4
**Overall Evaluation:** 3
**Confidence:** 3

**Weaknesses:**

1: Minor weaknesses that are easily fixable.

**Contributions Of The Paper:**

This paper describes SayNav which uses commonsense knowledge encoded in LLMs to generate high-level plans fro the multi-object navigation task. The novelty of this work is that it uses an LLM to ground relations between objects and the environment, and also to reason about those relationships to produce a plan.

**Ethical Considerations:**

(2) Poor: The paper fails to address crucial ethical considerations

**Nomination For Best Paper:**

Yes

**Questions For Authors:**

1. Can you provide more details on how the LLM is used to augment and refine the scene graph? Also, how is memory encoded in the graph?
2. Also, did you examine how success rate and SPL varied by the number of rooms or size of the house?

**Reproducibility:**

4: Authors promise to release code and domains (whichever apply).

**Strengths Of The Paper:**

The novel hierarchical planning approach based on using an LLM. Also the combination of a high-level plan with a low-level planner that executes the plan steps.

**Weaknesses Of The Paper:**

The method only achieves 60.32% success, which with only 3 objects to find, is not very impressive. It is not clear why their baseline performs so poorly given the optimal sequence of objects and their locations - it might be useful to compare against other methods that start with the same information in an unfamiliar environment.

---

> ### Author Rebuttal · Authors · 2024-01-28
>
> We thank the reviewer for the insightful comments.
>
> **Scene-graph Refinement:** LLM helps to identify the room-type based on perceived objects (see **L30-35 Appendix**). It also contributes to decision of an object’s classification as large or small object based on object class.
>
> **Memory:** Memory is encoded as metadata of room-nodes in the graph, which is updated once a room has been explored. We have also shown an alternative way of supporting memory in **Appendix Sec C** where the LLM itself keeps track of the rooms by being augmented with a conversational memory.
>
> **House Size:** We examined the variation of performance w.r.t. number of rooms and we observed similar performances across different sizes. We can account this behavior to the fact that LLM enables the agent to skip searching irrelevant rooms, significantly reducing the total number of steps required by the Low-Level Planner (LLP) to achieve the goal. In other words, the agent may end up searching only a few rooms to achieve the goal in a very large house.
>
> **Baseline:** Most existing LLM-based approaches for robot navigation operate in known and/or small-scale environments. However, there do exist RL / IL based approaches (ex: Gireesh N. et al. 2023, arXiv:2305.06178) and traditional SLAM-based approaches that can work in our problem setting. Many previous works (ex: Savva M. et al. 2019, arxiv:1904.01201) have shown that RL / IL based Point-goal Navigation (PointNav) outperforms SLAM-based approaches in unknown environments by a large extent (80% vs 62% in the mentioned example). Due to lack of open-source datasets / code from existing works, we chose to implement the strongest possible baseline agent which employs IL-based PointNav policy (see **L530-550**). However, one drawback with PointNav policy is its loss in performance in large-scale environments. As shown in **L500-508**, the PointNav policy performs better on short-range goals (98.5% SR, 0.93 SPL) than long-range goals (84.5% SR and 0.782 SPL). Although, the baseline has access to the locations of the goal objects, finding each object still requires it to plan a long-range path across multiple rooms. Note that we use the exact same LLP (PointNav) from the baseline agent for SayNav. However, SayNav addresses this shortcoming of PointNav policy by generating only short-range goals for LLP and hence, is able to beat the baseline.

---

### Meta-Review · Area_Chair_bzgn · 2024-02-04

**Recommendation:** Accept (Oral)
**Confidence:** 4

**Metareview:**

This paper proposes SayNav, a system for navigation tasks in unknown 3D environments, which uses a LLM to generate high-level plans which are then input to a low-level planner for navigation. SayNav is evaluated on the MultiON task, a task to look for specified objects in an unknown environment.
Although there was some divergence of opinion in the reviews prior to the rebuttal and discussion phase, the reviewers found the rebuttals to be convincing.

**Ethical Considerations:**

(1) Not Applicable: The paper does not have any ethical considerations to address